# Significant Enrichment of Potential Pathogenic Fungi in Soil Mediated by Flavonoids, Phenolic Acids, and Organic Acids

**DOI:** 10.3390/jof11020154

**Published:** 2025-02-17

**Authors:** Shaoguan Zhao, Yan Sun, Lanxi Su, Lin Yan, Xingjun Lin, Yuzhou Long, Ang Zhang, Qingyun Zhao

**Affiliations:** 1Spice and Beverage Research Institute, Chinese Academy of Tropical Agricultural Sciences/Key Laboratory of Genetic Resources Utilization of Aromatic Beverage Crops, Ministry of Agriculture and Rural Affairs/Hainan Key Laboratory of Genetic Improvement and Quality Control of Tropical Sweet and Spicy Beverage Crops, Wanning 571533, China; 2024203091@stu.njau.edu.cn (S.Z.);; 2Sanya Research Institute, Chinese Academy of Tropical Agricultural Sciences, Sanya 572025, China

**Keywords:** root exudates, soil microbial communities, potential pathogenic fungi, soil properties, microbial enrichment

## Abstract

It is well established that root exudates play a crucial role in shaping the assembly of plant rhizosphere microbial communities. Nonetheless, our understanding of how different types of exudates influence the abundance of potential pathogens in soil remains insufficient. Investigating the effects of root exudates on soil-dwelling pathogenic fungi is imperative for a comprehensive understanding of plant–fungal interactions within soil ecosystems and for maintaining soil health. This study aimed to elucidate the effects of the principal components of root exudates—flavonoids (FLA), phenolic acids (PA), and organic acids (OA)—on soil microbial communities and soil properties, as well as to investigate their mechanisms of action on soil potential pathogenic fungi. The results demonstrated that the addition of these components significantly modified the composition and diversity of soil microbial communities, with OA treatment notably altering the composition of dominant microbial taxa. Furthermore, the introduction of these substances facilitated the proliferation of saprophytic fungi. Additionally, the incorporation of flavonoids, phenolic acids, and organic acids led to an increased abundance of potential pathogenic fungi in the soil, particularly in the FLA and PA treatments. It was observed that the addition of these substances enhanced soil fertility, pH, and antioxidant enzyme activity. Specifically, FLA and PA treatments reduced the abundance of dominant microbial taxa, whereas OA treatment altered the composition of these taxa. These findings suggest that the inclusion of flavonoids, phenolic acids, and organic acids could potentially augment the enrichment of soil potential pathogenic fungi by modulating soil properties and enzymatic activities. These results offer valuable insights into the interactions between plants and fungal communities in soil ecosystems and provide a scientific foundation for the management and maintenance of soil health.

## 1. Introduction

Soil-borne phytopathogens play a critical role in the development of plant diseases, which, in turn, have profound implications for the global agricultural economy [1]. Notably, fungi such as *Fusarium graminearum*, *Aspergillus sojae*, *Penicillium italicum*, and *Talaromyces macrosporus* are frequently encountered as pathogens responsible for wheat Fusarium head blight, soybean root rot, and citrus fruit rot, respectively [2,3,4]. An in-depth understanding of the proliferation of these pathogenic fungi in soil is essential for mitigating the incidence of soil-borne plant diseases, thereby ensuring food security and minimizing crop losses. The concept of pathogen enrichment in soil refers to the increase or accumulation of specific pathogenic fungal populations within the soil matrix. Previous studies have elucidated several factors that contribute to the significant enrichment of soil-borne pathogens. These include the accumulation of plant residues, the effects of continuous cropping, soil physicochemical properties, prevailing environmental conditions, and suboptimal agronomic practices [5,6]. Each of these factors plays a role in creating an environment conducive to the proliferation of soil-borne pathogens. In addition, root exudates, which serve as a primary carbon source for soil microorganisms, are crucial in the complex interactions between soil microbial communities and plants [7]. Although extensive research has demonstrated that root exudates generally promote beneficial microbial activity, enhance rhizosphere conditions, and foster plant–microbe interactions [8,9], our current knowledge of the specific effects of various types of root exudates on pathogenic fungi remains limited.

Different types of substances secreted by plant roots have selective effects on the composition and functionality of soil microbial communities [10]. Flavonoids, as secondary metabolites, have been extensively studied and are known to possess various physiological functions. For instance, flavonoids exhibit antioxidant properties, enhance nutrient uptake in plants, and mitigate heavy metal toxicity in soils [11]. However, some studies suggest that flavonoids may exert stimulatory effects on fungal pathogens, leading to the increased abundance and activity of these pathogens, thereby competing with mycorrhizal fungi and potentially stabilizing mycorrhizal fungal infections [12]. Similarly, phenolic compounds, which are also secondary metabolites produced by plant roots, are frequently considered primary contributors to soil autotoxicity [13]. For example, cinnamic acid has been shown to enhance the pathogenicity of *Fusarium* cells by stimulating them to produce enzymes that play a notable role in infecting the host and inhibiting the defense response of leguminous roots, thus promoting the infection of legumes with root rot [14].

The interaction between organic acid compounds and soil microorganisms holds significant importance. Research has established a clear correlation between organic acids and the occurrence of wheat root rot disease, specifically noting a substantial production of succinic acid associated with this ailment [15]. Additionally, acidic substances added to the soil can decrease its pH, consequently fostering an environment conducive to the proliferation of pathogenic fungi [16]. These observations indicate that flavonoids, phenolic acids, and organic acids, while serving vital physiological roles in plants, may inadvertently stimulate soil-borne fungal pathogens through their interactions with soil microorganisms. This, in turn, can profoundly affect plant growth and vitality [12,17,18]. Numerous common pathogenic fungal species reside in soil, including *Fusarium* spp. causing wilts, *Phytophthora* spp. and *Aspergillus* spp. inducing root rots, and *Penicillium* spp. and *Talaromyces* spp. leading to fruit rots [19]. Typically, these pathogenic fungi initiate the infection process by attaching to the plant surface through secreted adhesive materials or specialized structures upon contacting the soil. They then secrete enzymes to degrade the plant’s cell walls, enabling deeper penetration into the plant tissue. Once inside, they exploit the host’s nutrients for reproduction and secrete toxins or hormones, ultimately causing cellular damage, hormonal imbalances, and cell death in the host plant [20,21]. Flavonoids, phenolic acids, and organic acids have the potential to modify the soil microenvironment, thereby enhancing the growth conditions for pathogenic fungi and supplying them with additional nutrients. This, in turn, can facilitate their invasion into plants. Specifically, certain phenolic acids and organic acids are recognized as autotoxic substances in soil, potentially diminishing the diversity of soil microorganisms. A decrease in community diversity may suggest a reduction in ecosystem resilience, subsequently elevating the plants’ susceptibility to pathogenic fungal proliferation and disease outbreaks [22,23].

Indeed, it is crucial to gain a deeper understanding of the mechanisms underlying the effects of flavonoids, phenolic acids, and organic acids on soil microorganisms and pathogenic fungi in order to better control and prevent plant diseases. This will help protect the healthy growth of plants and maintain the stability of soil ecosystems. In this work, we have chosen to add the major flavonoids, phenolic acids, and organic acids present in root exudates to the soil and observe changes in soil characteristics and the dominant fungal genera. Our research aims to (i) investigate the impact of these substances on soil properties, (ii) assess fungal community diversity through high-throughput sequencing technologies, and (iii) compare the abundance and composition of major pathogenic fungal genera in the soil under the influence of these exogenous substances. Through these experiments, we hope to further elucidate the mechanisms by which flavonoids, phenolic acids, and organic acids influence soil microorganisms and pathogenic fungi, providing a more scientifically grounded basis for plant health management and disease control.

## 2. Materials and Methods

### 2.1. Experimental Site

The study took place in the plantation resource garden of the Spice and Beverage Research Institute located in Hainan Province, China, at coordinates 110°6′ E longitude and 18°24′ N latitude. The soil at the experimental site had an organic carbon content of 164.2 g/kg, an available nitrogen content of 98.5 g/kg, an available phosphorus content of 250.2 mg/kg, and an available potassium content of 84.1 mg/kg.

### 2.2. Experimental Materials

The soil samples were obtained from a coffee plantation. During the sample collection process, the top layer of soil was uncovered, and plant residues and large particles were eliminated. Soil was excavated from a depth of 0–20 cm and adequately homogenized by passing it through a sieve with a mesh size of less than 2 mm to eliminate smaller plant residues and particles, ensuring its suitability for experimental purposes.

The chosen additives were primarily composed of flavonoids, phenolic acids, and organic acids, which are the main constituents of root exudates. The additives were analytically pure reagents purchased from reagent companies (Aladdin Reagents, Los Angels, CA, USA).

### 2.3. Experimental Design

The experiment was designed to assess the influence of root exudates on soil microbial communities using a randomized block design with four treatments: a control (CK) consisting of a 20% sterile methanol solution, and three treatments with the same solution supplemented with organic acids (OA), phenolic acids (PA), and flavonoids (FLA), respectively. Each treatment was replicated six times (Appendix A). To investigate whether 20% sterile methanol solution would affect the diversity and structural composition of soil microbial communities, a preliminary experiment was conducted. The results showed that the microbial community diversity and structural composition in treated soil (KCK) were not significantly different from those in the control (Appendix A). A 60 g aliquot of soil was placed in large containers for each replicate. The soil was then positioned in a coffee field to standardize environmental conditions. Daily, exudate solutions were neutralized to a neutral pH using dilute hydrochloric acid and ammonia, acknowledging the pivotal role of soil pH in dictating soil microbial community structure. The neutralized solutions were applied to the soil at a rate of 0.075 mgC/g/d, as per established protocols [24,25,26]. The carbon content from each carbon source in the simulated root exudates was equilibrated, with citric acid, succinic acid, and tartaric acid each comprising one-third of the total carbon content, excluding methanol-derived carbon. After a 6-week incubation period, destructive sampling was conducted, and samples were cryopreserved at −80 °C for subsequent analysis.

### 2.4. Analysis of Soil Physico-Chemical Properties and Soil Enzyme Activities

Soil pH was quantified using a FE28 pH meter (Mettler Toledo, Shanghai, China) with a soil-to-water ratio of 1:2.5. Alkali–nitrogen content was ascertained via the diffusion method. The alkali-hydrolyzed nitrogen content was evaluated employing a Multi-N/C3100 total organic carbon analyzer (Analytik Jena, Jena, Germany). Soil-available potassium (AK) was measured by flame photometry using an AK 6400A instrument (Changsha, China). Soil-available phosphorus (AP) was determined following Bray’s method with a UV2310 II spectrophotometer (Shanghai, China). Enzymatic activities, including soil urease (S-UE), soil dehydrogenase (S-DHA), soil cellulase (S-CL), soil catalase (S-CAT), peroxidase (S-POD), soil polyphenol oxidase (S-PPO), soil acid phosphatase (S-ACP), and soil alkaline phosphatase (S-ALP), were assayed using micromethod kits from Suzhou Grace Biotechnology Co., Ltd. (Grace Biotechnology, Suzhou, China), and quantified with a SynergyH1 enzyme-linked immunosorbent assay (ELISA) reader (Agilent Technologies, Santa Clara, CA, USA).

### 2.5. Soil DNA Extraction and Sequencing

Total soil DNA was extracted and purified using commercial soil DNA extraction kits. DNA concentration was measured using a Nanodrop spectrophotometer (Thermo Fisher Scientific, Wilmington, DE, USA), and the integrity of the DNA was confirmed by agarose gel electrophoresis on a 1.2% gel [27]. For fungal community analysis, the internal transcribed spacer (ITS) regions were amplified with primers ITS5F (5′-GGAAGTAAAAGTCGTAACAAGG-3′) and ITS1R (5′-GCTGCGTTCTTCATCGATGC-3′), incorporating barcode sequences specific to each sample. PCR amplification targeted variable regions within the ITS gene fragments [28]. The resulting PCR products were quantified by fluorescence using the Quant-iT PicoGreen dsDNA Assay Kit and a Microplate reader (BioTek Instruments, Winooski, VT, USA, FLx800). Samples were pooled in equimolar ratios based on sequencing volume requirements, as determined by fluorescence quantification. Sequencing libraries were constructed using the Illumina TruSeq Nano DNA LT Library Prep Kit (Illumina, San Diego, CA, USA). The libraries were subjected to paired-end sequencing on the Illumina platform to analyze the community-derived DNA fragments.

### 2.6. Bioinformatics Analysis

Bioinformatics Analysis: The microbiome data analysis was conducted using QIIME 2 version 2019.4 [29], with minor adjustments based on the official tutorials available in the QIIME 2 documentation.

Data Processing: Raw sequencing data underwent demultiplexing with the demux plugin, followed by primer trimming using the cutadapt plugin [30]. Subsequent steps included the quality filtering, denoising, merging, and chimera removal of the sequences, all performed with the DADA2 plugin [31]. Amplicon sequence variants (ASVs) that were not singletons were aligned using mafft and a phylogenetic tree was constructed with FastTree2 [32].

Taxonomy Assignment and Diversity Metrics: Taxonomy was assigned to the ASVs using the classify-sklearn naïve Bayes taxonomy classifier within the feature-classifier plugin [33], against the Greengenes 13_8 99% OTUs reference sequences [34]. Diversity indices, including alpha-diversity metrics (Chao1, Observed species, Shannon, Simpson, Faith’s PD) and beta-diversity metrics (weighted and unweighted UniFrac [35,36], Jaccard distance, and Bray–Curtis dissimilarity), were calculated using the diversity plugin. Samples were rarefied to an equal number of sequences per sample prior to these estimates.

### 2.7. Statistical Analysis

All statistical analyses were performed in R 4.2.0 and visualized with ggplot2. Principal Coordinate Analysis (PCoA) based on Bray–Curtis distance was used to explore differences in several root secretions on fungal community structure, respectively. Soil physicochemical properties, soil enzyme activity, soil microbial diversity indices, the relative abundance of specific microbial taxa, and the relative abundance of soil microbial functional groups were analyzed using one-way ANOVA to determine differences in soil property indices among the addition of different substances. The statistical significance (*p* < 0.05) was calculated using the Tukey test. Correlations between soil properties, soil enzyme activity, and soil microbial community diversity were calculated and analyzed using Spearman correlation matrices. FUNGuild (v1.0) was used to analyze and predict microbial community function, respectively.

## 3. Results

### 3.1. The Effects of Exogenous Material Addition on Soil Factors

The intake of three types of exogenous substances altered the properties of soil (Table 1). The addition of flavonoids significantly increased soil organic matter, available potassium, alkaline nitrogen content, and pH by 36.6%, 29.7%, 9.4%, and 18.9%, respectively, while significantly decreasing the available potassium content by 40.3%. The incorporation of organic acids significantly increased soil organic matter, available potassium, alkaline nitrogen content, and pH by 42.8%, 32.5%, 5.3%, and 66.6%, respectively, while significantly decreasing the available potassium content by 37.7%. The addition of phenolic acids significantly increased soil organic matter, available potassium, alkaline nitrogen content, and pH by 19.3%, 34.6%, 18.0%, and 44.2%, respectively, while significantly decreasing the available potassium content by 58.0% (*p* < 0.05).

The addition of flavonoids significantly increased S-CAT and S-ALP activity by 15.3% and 88.0%, respectively. The incorporation of organic acids significantly increased S-PPO, S-CL, S-UE, S-CAT, S-POD, S-DHA, and S-ALP activity in soil by 144.1%, 195.9%, 19.9%, 56.1%, 43.7%, 97.3%, and 72.2%, respectively. The addition of phenolic acids significantly increased S-PPO, S-CAT, S-POD, and S-DHA activity in soil by 31.4%, 37.7%, 54.7%, and 61.1%, respectively (Table 2, *p* < 0.05).

### 3.2. The Influence of Exogenous Substances on the Structure of Soil Fungal Communities

In order to investigate the effect of external substances on soil microbial communities, fungal communities were analyzed under different treatments, and the composition of fungal communities was influenced by external substances (Figure 1). The Venn diagram shows that 74 OUTs are shared across all treatments, while CK, FLA, OA, and PA have 1211, 281, 703, and 227 unique OUTs, respectively (Figure 1a). Among the three substances, flavonoids and phenolic acids significantly reduced the Shannon and Chao1 indices of soil fungal communities, indicating a significant decrease in richness and diversity relative to CK (Figure 1b). Furthermore, the PCoA score plot shows that flavonoids and phenolic acids change the structure of soil fungal communities relative to CK (Figure 1c). Based on the analysis of fungal community composition at the genus level, *Fusarium* (11.0%), *Saitozyma* (21.0%), *Mortierella* (10.3%), *Humicola* (2.2%), *Trichoderma* (1.2%), and *Penicillium* (1.6%) were the dominant genera (relative abundance > 1%) among the 304 genus-level fungi identified, except for those that were unidentified. The addition of the three substances significantly increased the relative abundance of *Fusarium*, while the relative abundance of other dominant genera decreased to varying degrees (Appendix A). Additionally, the dominant genera in the soil fungal community were altered (Figure 1d).

### 3.3. Effects of Foreign Substances on Potential Fungal Pathogens

A clustered analysis of fungal community composition revealed distinct taxonomic relationships among 15 fungal genera across treatments (Figure 2a). Compared to CK, shifts in dominance were observed, with specific genera (e.g., *Aspergillus*, *Fusarium*, *Penicillium*, and *Trichoderma*) exhibiting differential distribution patterns in response to treatments (Figure 2b). Notably, the FLA treatment induced a pronounced increase in the relative abundance of *Talaromyces* (Figure 2c). While these genera include documented soil-borne pathogens, their distribution varied significantly among treatments: flavonoid (FLA) and phenolic acid (PA) treatments showed a stronger clustering of *Aspergillus* and *Fusarium*, whereas the polysaccharide (PA) treatment favored *Penicillium* and *Trichoderma* (Figure 2b). A ternary analysis further highlighted treatment-specific enrichment trends, with FLA and PA driving the most distinct genus-level redistributions (Figure 2b).

Compared to CK, the relative abundance of *Aspergillus* increased significantly by 1048.7% in the FLA treatment. The relative abundance of *Fusarium* increased significantly by 821.8%, 938.9%, and 808.4% in the FLA, OA, and PA treatments, respectively. The change in the relative abundance of *Penicillium* did not reach a significant level in the FLA, OA, and PA treatments. The relative abundance of *Talaromyces* increased significantly by 5033.9% in the FLA treatment (Figure 2c). It can be seen that the addition of external substances resulted in a high degree of enrichment of these pathogenic fungal genera (*p* < 0.05).

### 3.4. Relationship Between Soil Properties and Fungal Communities and Potential Pathogenic Fungi

According to the Mantel distance algorithm (Figure 3a), significant positive correlations were observed between the relative abundance changes in *Fusarium* and S-CL (R = 0.17), S-UE (R = 0.40), S-POD (R = 0.21), and S-ACP (R = 0.32). Likewise, the relative abundance changes in *Talaromyces* showed a significant positive correlation with S-ALP (R = 0.28), whereas the relative abundance changes in *Penicillium* exhibited a significant positive correlation with S-CAT (R = 0.15). Based on the Pearson distance algorithm, S-PPO showed significant positive correlations with S-DHA (R = 0.54, *p* < 0.05), S-POD (R = 0.72, *p* < 0.05), S-CAT (R = 0.80, *p* < 0.05), and S-CL (R = 0.47, *p* < 0.05). Significant negative correlations were found between S-CL and S-UE (R = 0.49, *p* < 0.05), as well as between S-CL and S-DHA (R = 0.75, *p* < 0.05) and S-ACP (R = 0.69, *p* < 0.05). S-UE also showed significant negative correlations with S-ACP (R = 0.63, *p* < 0.05). Moreover, S-CAT exhibited significant positive correlations with S-POD (R = 0.52, *p* < 0.05) and S-DHA (R = 0.61, *p* < 0.05). In terms of soil physicochemical factors, only soil AHN content showed a significant positive correlation with the relative abundance changes in *Fusarium*. In addition, SOM and AHN showed significant positive correlations with AK (R = 0.73 and R = 0.41, *p* < 0.05, respectively), and a significant negative correlation with AP (R = 0.58 *p* < 0.05). Furthermore, AK showed a significant negative correlation with AP (R = 0.44, *p* < 0.05), while AP exhibited a significant negative correlation with AHN (R = 0.51, *p* < 0.05).

The abundance changes of 20 major fungal genera were found to be associated with changes in soil factors (Appendix A). The RDA ordination analysis indicated that the soil factors with significant effects included S-ALP (R^2^ = 0.35, *p* < 0.01), S-UE (R^2^ = 0.71, *p* < 0.01), S-POD (R^2^ = 0.29, *p* < 0.05), S-ACP (R^2^ = 0.30, *p* < 0.05), S-CL (R^2^ = 0.36, *p* < 0.05), and AHN (R^2^ = 0.52, *p* < 0.01). Specifically, S-ALP and S-UE showed a positive correlation with *Fusarium*, while S-CL, S-ACP, S-POD, and AHN demonstrated a negative correlation with *Fusarium*. For *Talaromyces*, significant positive correlations were observed with S-ACP, S-ALP, and S-UE, while significant negative correlations were found with AHN, S-CL, and S-POD. Moreover, S-POD exhibited negative correlations with *Aspergillus* and *Penicillium*, while S-ALP, S-ACP, S-UE, S-CL, and AHN showed negative correlations with *Aspergillus* and *Penicillium* (Figure 3b,c).

### 3.5. The Addition of Exogenous Substances Resulted in Changes in the Structure of Functional Groups

The functional prediction of fungal communities under various treatments was illustrated (Figure 4a). The dendrogram revealed substantial alterations in fungal functional groups in samples treated with FLA, OA, and PA, in comparison to the control (CK). Nonetheless, the response of soil fungal functional communities to the different substances appeared relatively homogeneous. Furthermore, Principal Coordinate Analysis (PCoA) demonstrated a significant modification in the structure of fungal functional groups following the addition of carbon-source substances, relative to CK (Figure 4b). Specifically, in the OA treatment, the abundance of all three functional group categories exhibited significant increases compared to CK: a 118.0% increase in saprotrophic functional group abundance, a 158.0% increase in symbiotic functional group abundance, and a 171.0% increase in pathogenic functional group abundance. Increases in the abundance of saprotrophic functional groups were also observed in FLA and PA treatments, with increases of 192.0% and 246.0%, respectively. The abundance of pathogenic functional communities in the FLA treatment was significantly enhanced by 136.0% (Figure 4c).

## 4. Discussion

### 4.1. Additives Reduce the Diversity of Soil Fungal Communities

Soil microbial diversity plays a crucial role in maintaining the balance and functionality of soil ecosystems [37]. A diverse array of microorganisms enhances nutrient cycling efficiency, augments soil resilience against diseases and pests, and promotes plant growth and overall health [38,39]. Prior research has demonstrated that specific compounds secreted by plant roots can accumulate in the soil, adversely affecting both soil microbial diversity and plant health. These compounds are termed autotoxic compounds [18]. For instance, phenolic compounds such as phenol, 2,4-di-tert-butylphenol, and vanillic acid are frequently regarded as autotoxic substances, as their accumulation can disrupt the soil microbial community and diminish microbial diversity. Conversely, fungi are heterotrophic organisms that depend on external carbon sources, with various fungal taxa employing distinct carbon substrates [40]. Certain flavonoids and organic acids found in root exudates, including lactic acid, oxalic acid, and citric acid, can only be mineralized by specific microorganisms, while the majority of soil microbial populations are unable to utilize these substrates, thereby influencing microbial diversity in the soil [41,42]. These findings align with the results of the present study, wherein the Shannon index and Chao1 index revealed a significant reduction in fungal community diversity in the soil treated with FLA and PA.

In this study, the application of FLA and PA treatments resulted in a substantial increase in soil nutrient levels. Prior research indicates that root exudates can stimulate the growth and activity of specific soil microorganisms by serving as a source of energy and nutrients [43]. Consequently, the introduction of flavonoids and phenolic acids may selectively influence the fungal community in the soil, leading to an augmentation of certain dominant microbial species while suppressing others [9,44,45]. This effect is corroborated by changes observed in fungal operational taxonomic units (OTUs). Specifically, the addition of flavonoids and phenolic acids resulted in a reduction in the number of fungal OTUs, with a notable decrease in the number of unique OTUs. Additionally, flavonoids may interact with microbial cell surfaces, potentially disrupting microbial communication or signal transduction pathways, thereby affecting microbial interactions and overall diversity [46,47]. Conversely, flavonoids and phenolic acids (e.g., salicylic acid and naringenin) can exert inhibitory effects on specific microorganisms by modulating metabolic pathways, compromising cell-membrane integrity, and altering membrane permeability [48,49], thereby directly impeding fungal growth and reproduction [50,51]. This may result in the enrichment of fungal species that are either tolerant to or unaffected by flavonoids and phenolic acids. Thus, the observed reduction in the number of unique OTUs may be a significant factor contributing to the decreased fungal community diversity.

### 4.2. The Structural Composition of Soil Fungi Genera and Functional Communities Was Changed by the Addition

Root exudates function as a vital carbon source for rhizosphere microorganisms, and specific compounds within these exudates demonstrate selective toxicity. This implies that they are detrimental to certain microbial species while exerting minimal impact on others [8]. This circumstance gives rise to an imbalance in carbon-source utilization among rhizosphere microorganisms, ultimately leading to alterations in the abundance of specific microbial species. Such changes have the potential to induce modifications in the structure of soil microbial communities [9,14,52]. In the present study, the introduction of FLA and PA notably altered the configuration of soil microorganisms. Upon analyzing the foremost 20 fungal genera across all treatment groups, discernible variations in the composition and prevalence of dominant species were observed when compared to the CK group. The incorporation of flavonoids and phenolic acids resulted in a diminution of dominant species abundance. Conversely, the addition of organic acids, while not affecting the abundance of dominant species, induced shifts in their composition. These three compound types also exhibited significant disparities in their capacity to modify fungal functional groups when compared to CK. The findings revealed that the introduction of these compounds led to a decrease in the abundance of unclassified functional groups. Simultaneously, there was an augmentation in fungal functional groups characterized by pathotroph–saprotroph–symbiotroph and saprotrophic nutrition. This provides indirect evidence for the selective enrichment of fungal species by various types of root exudates [10]. Furthermore, flavonoids and phenolic acids were found to alter the activity of soil microorganisms [11,53]. Organic acids possess the capability to dissolve soil particles, facilitate mineral dissolution and dissociation, and liberate organic and inorganic nutrients for microbial utilization [54]. These observations align with the findings of the current study, which demonstrated that the addition of exogenous substances significantly elevated the activity of specific soil enzymes (such as phosphatases and antioxidant enzymes) and augmented the content of alkaline hydrolyzable nitrogen. These changes significantly influenced the composition of soil microorganisms.

### 4.3. The Enrichment Mechanisms of Major Pathogenic Fungus Genera

In soil ecosystems, a diverse array of potential pathogenic fungi exists, which can instigate various plant diseases and adversely affect the health and yield of crops and horticultural plants [55]. The present study identified several of these potential pathogenic fungi among the dominant fungal species, including *Fusarium* [19], *Aspergillus* [21], *Penicillium* [56], and *Talaromyces* [57]. Further analysis revealed that the response of these pathogenic fungi to key components in root exudates led to their increased abundance in soil upon the addition of exogenous materials. Specifically, in the FLA and PA treatments, there was a significant reduction in the abundance of other dominant fungi, which corresponded with a notable enhancement in the presence of pathogenic fungi. This observation is in contrast with previous findings. Prior research has indicated that flavonoids typically inhibit the growth of plant pathogenic fungi [58], while phenolic acids generally promote their growth [6,14]. Such discrepancies may be attributed to the impact of added exogenous materials on soil properties. It has been demonstrated that root exudates can lead to the degradation of fixed soil nutrients, such as organic nitrogen and phosphorus [59,60]. In the current study, a significant positive correlation was observed between the content of alkaline-hydrolyzable nitrogen in the soil and the relative abundance of *Fusarium*.

Root exudates not only supply energy and nutrients to the soil but also stimulate the growth and activity of soil microorganisms. Studies have indicated that increased enzyme activity can serve as a proxy for microbial and other biological activity within the soil ecosystem [61]. In comparison to the control (CK) treatment, the addition of three distinct substances led to varying degrees of enhancement in soil enzyme activity, with organic acids demonstrating the most pronounced effect. Previous research has posited that the escalation of antioxidant enzyme activity in the soil may be associated with the proliferation of pathogenic bacteria [62]. Antioxidant enzymes are capable of mitigating oxidative stress induced by pathogenic bacteria, thereby reducing oxidative damage and inhibiting fungus growth [63]. In this study, it was observed that all three treatments significantly elevated the activity of antioxidant enzymes, including polyphenol oxidase, catalase, peroxidase, and dehydrogenase. This suggests that the incorporation of these substances may indirectly activate defense mechanisms in the soil. The increased activity of these antioxidant enzymes is likely attributable to the higher abundance of pathogenic fungi present in the soil. Saprophytic fungi, which decompose organic materials to extract energy and nutrients, play a crucial role in this process [64]. During decomposition, saprophytic fungi produce a suite of enzymes capable of breaking down complex compounds in plant residues, such as cellulose, lignin, and proteins [65]. The observed increase in cellulase and phosphatase activity may be indicative of an augmented functional group of saprophytic fungi, with these enzyme activities showing significant positive correlations with the relative abundance of *Fusarium* and *Talaromyces*. These enzymes degrade plant residues into smaller organic molecules, some of which may serve as nutrient sources for pathogenic fungi, thus providing essential carbon and other nutrients for their growth. Consequently, the decompositional activities of saprophytic fungi may facilitate the proliferation and enrichment of pathogenic fungi.

## 5. Conclusions

The addition of flavonoids and phenolic acids in root exudates can reduce the diversity of fungal communities and decrease the abundance of some dominant fungi while increasing the enrichment of potential pathogenic fungi in the soil. In contrast, the addition of organic acids may alter the composition of dominant fungi. The addition of exogenous materials can alter the structure of soil microorganisms, reducing or changing the abundance of dominant microbial populations. The addition of flavonoids and phenolic acids significantly enriches the potential pathogenic fungi in the soil and is accompanied by increased enzyme activity. Moreover, this addition can also increase the activity of saprophytic functional groups, particularly the increase in cellulase and phosphatase activity, which is related to the increase in saprophytic fungi. The decomposition function of saprophytic fungi may provide a nutrient source for pathogenic fungi, promoting their proliferation and enrichment.

In conclusion, the addition of flavonoids, phenolic acids, and organic acids in root exudates has significant impacts on soil microbial diversity, the enrichment of pathogenic fungi, soil enzyme activity, and saprophytic functional groups. These findings are of great significance for a deeper understanding of the interactions between rhizosphere microorganisms and soil ecosystems, as well as their impacts on soil health and crop production.

## Figures and Tables

**Figure 1 jof-11-00154-f001:**
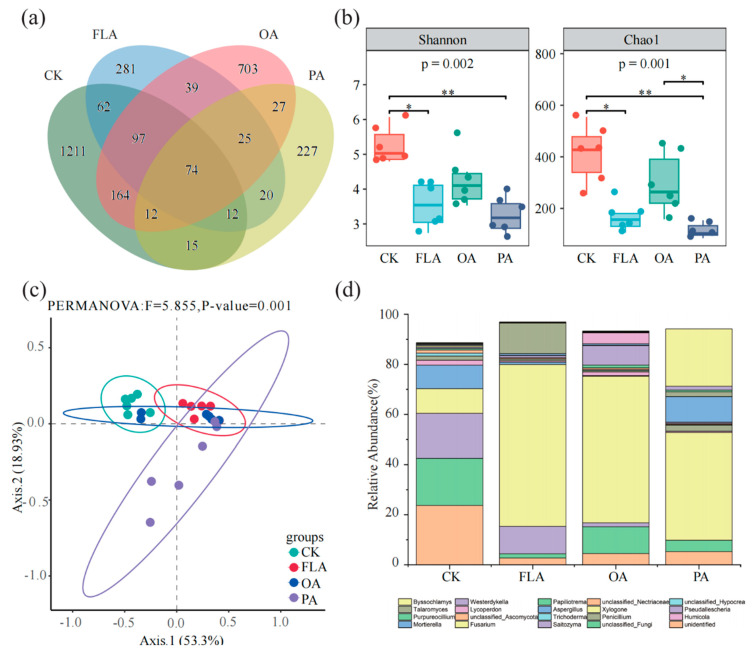
Based on the OTU level, the composition of fungal community in different treatments was analyzed. (**a**) The Venn diagram of different treatments was constructed according to the OTU number. (**b**) The difference in fungal community diversity among different treatments was analyzed based on the OTU level. (**c**) PCoA analysis was conducted to analyze the composition of fungal communities in different treatments. (**d**) The main classification of fungal communities at the genus level in different treatments was analyzed. “*” means *p* < 0.05 and “**” means *p* < 0.01 by Kruskal–Wallis rank-sum test. CK: 20% sterile methanol solution, FLA: flavonoids; OA: organic acids, PA: phenolic acids.

**Figure 2 jof-11-00154-f002:**
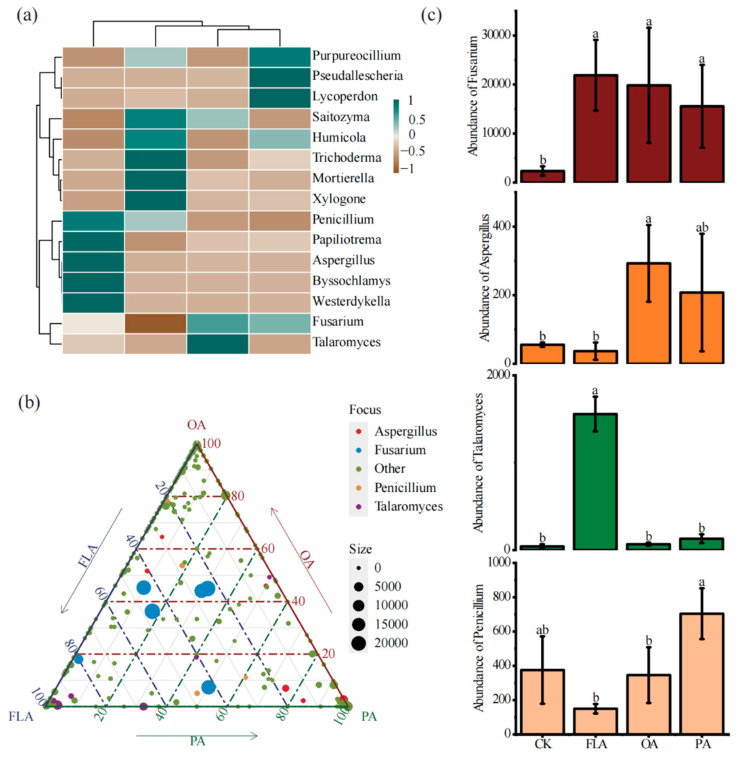
The genus level composition of fungal communities in different treatments was analyzed using heatmap visualization and ternary analysis. (**a**) The top 20 genera of each treatment were depicted in the heatmap. (**b**) Ternary analysis was also employed to evaluate and compare the fungal communities between different treatments at the genus level. (**c**) Significant changes in four major pathogenic fungal genera across different treatments. Different small letters represent significant differences between different treatments. CK: 20% sterile methanol solution, FLA: flavonoids; OA: organic acids, PA: phenolic acids.

**Figure 3 jof-11-00154-f003:**
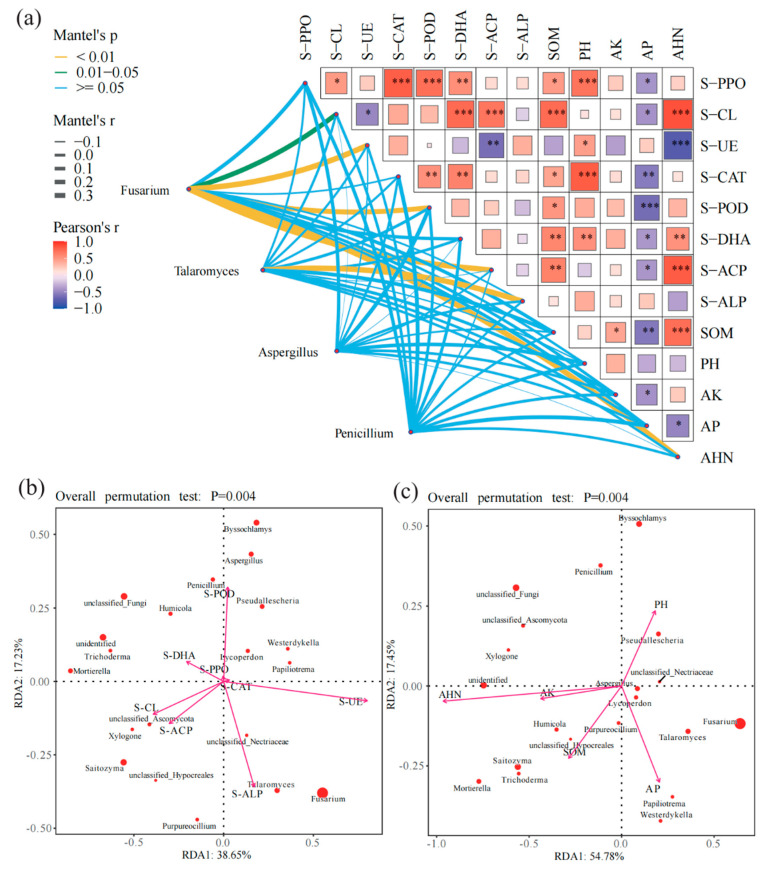
Changes in the main pathogenic fungi genera in fungal communities under different treatments and their relationship with soil factors. (**a**) Network heatmap showing the association between four major pathogenic fungal genera and soil enzyme activity. (**b**,**c**) are redundancy analysis (RDA) plots, showing the distribution of fungal genera and chemical indices on the RDA1 and RDA2 axes. Arrows represent the direction and intensity of chemical indices, dots represent different fungal genera, and the significance level is *p* = 0.004. “*” indicates *p* < 0.05, “**” indicates *p* < 0.01, and “***” indicates *p* < 0.001, for both the Pearson correlation and Mantel test.

**Figure 4 jof-11-00154-f004:**
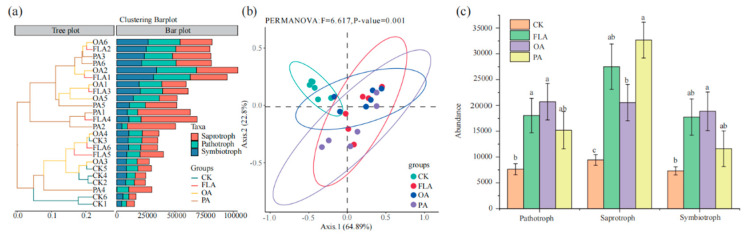
Functional predictions of fungal communities under different treatments were analyzed. (**a**) Clustering analysis was used to determine the functional groups in fungal communities under different treatments. (**b**) Principal Coordinate Analysis (PCoA) was used to evaluate the functional diversity and composition of fungal communities under different treatments. (**c**) The differences among the three functional groups across different treatments. Different small letters represent significant differences between different treatments. CK: 20% sterile methanol solution, FLA: flavonoids, OA: organic acids, PA: phenolic acids.

**Table 1 jof-11-00154-t001:** Effects of different root secretions on soil physico-chemical properties.

Treatments	SOM (g/kg)	pH	AK (mg/kg)	AP (mg/kg)	AHN (mg/kg)
CK	169.2 ± 8.6 d	4.7 ± 0.1 d	75.4 ± 0.8 b	300.8 ± 35.3 a	92.0 ±1.4 d
FLA	231.0 ± 1.9 b	5.6 ± 0.1 c	97.8 ± 5.3 a	179.6 ± 38.1 b	100.6 ± 0.9 b
OA	241.6 ± 12.2 a	7.8 ± 0.1 a	99.9 ± 2.0 a	187.3 ± 19.7 b	96.8 ± 3.2 c
PA	201.8 ± 1.3 c	6.8 ± 0.1 b	101.5 ± 4.2 a	126.4 ± 3.6 c	108.5 ± 1.7 a

Note: SOM (g/kg): soil organic carbon; pH: a measure of the acidity or alkalinity; AK (mg/kg): soil-available potassium; AP (mg/kg): soil-available phosphorus; AHN (mg/kg): alkali-hydrolyzed nitrogen. Different letters indicate significant differences between treatments (*p* < 0.05). n = 3. CK: 20% sterile methanol solution, FLA: flavonoids; OA: organic acids, PA: phenolic acids. Different letters indicate significant differences between treatments under the same soil physicochemical properties.

**Table 2 jof-11-00154-t002:** Effects of different root secretions on soil enzyme activities.

Treatments	S-PPO nmol/h	S-CL μg/d/g	S-UE μg/d/g	S-CAT μmol/h/g	S-POD nmol/h/g	S-DHA μg/d/g	S-ACP nmol/h/g	S-ALP nmol/h/g
CK	212.6 ± 9.2 c	210.8 ± 47.7 b	647.0 ± 40.3 b	273.0 ± 13.1 d	1808.9 ± 92.8 b	1216.0 ± 79.9 c	1514.2 ± 205.1 a	811.6 ± 59.0 b
FLA	228.8 ± 40.0 c	162.4 ± 49.4 b	670.4 ± 19.3 b	314.9 ± 27.2 c	2171.1 ± 49.2 b	1299.5 ± 317.6 c	1451.6 ± 178.2 a	1525.7 ± 144.0 a
OA	518.9 ± 50.9 a	623.8 ± 28.9 a	775.8 ± 22.6 a	426.2 ± 0.8 a	2599.8 ± 631.3 a	2398.7 ± 210.2 a	1245.8 ± 329.7 a	1397.6 ± 652.7 a
PA	279.23 ± 20.9 b	174.5 ± 9.1 b	636.9 ± 2.2 b	376 ± 6.2 b	2797.6 ± 61.3 a	1959.3 ± 20.9 b	1581.3 ± 279.4 a	586.1 ± 127.5 b

Note: S-PPO (nmol/h/g): soil polyphenol oxidase; S-CL (μg/d/g): soil cellulase; S-UE (μg/d/g): soil urease; S-CAT (μmol/h/g): soil catalase; S-POD (nmol/h/g): soil peroxidase; S-DHA (μg/d/g): soil dehydrogenase; S-ACP (nmol/h/g): soil acid phosphatase; S-ALP (nmol/h/g): soil alkaline phosphatase. Different letters indicate significant differences between treatments under the same soil enzyme activity, n = 3, treatments abbreviations and indicator abbreviations see Table 1.

## Data Availability

The data presented in this study are openly available in FigShare at https://doi.org/10.6084/m9.figshare.27985826.v1, reference number 27985826.

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
