# Peer review of "Significant Enrichment of Potential Pathogenic Fungi in Soil Mediated by Flavonoids, Phenolic Acids, and Organic Acids"

_jof, 2025, doi:10.3390/jof11020154_

Round 1
Reviewer 1 Report (Previous Reviewer 2)
I consider the work to be relevant and the results very interesting, especially because the addition of substances exuded by plant roots causes an increase and significant alteration of fungal pathogens. The findings reported are undoubtedly relevant and raise the need for more in-depth research to understand the mechanisms involved in these phenomena.
I acknowledge the effort made to improve the manuscript since the previous version did not allow for an adequate analysis.
In the "detailed comments" section I present observations that, in my opinion, should be made to the new version of the document in order to be published.
Below I mention my observations.
· In section 2.1 please use abbreviations such as kg, g, mg.
· In section 2.2 include information on the origin of the additives used. In section 2.6 I suggest omitting the paragraph from lines 161 to 163 which talks about the link to the QIIME 2 documentation.
· On line 168 correct fasttree2 to FastTree2
· On lines 184 and 185 correct "Funguild were....." to "FUNGuild was...."
· In the paragraph from lines 232 to 239, the results of the effect of the treatments on the correlation of fungal and bacterial communities are mentioned and reference is made to figures 2a, 2b and 2c to show the increase in the relative abundance of bacterial genera (figure 2a), the significant increase in pathogenic bacteria in two of the treatments (figure 2b) and the increase in the relative abundance of the Talaromyces genus in the FLA treatment (figure 2c), however, the figures only show the taxonomic relationship of 15 fungal genera (figure 2a), the distribution of 4 genera of fungi in the different treatments (figure 2b) and the abundance of 4 genera of fungi in the different treatments (figure 2c). Therefore, the description of the results in the paragraph does not match the figures to which they refer. There is no clarity between the paragraph and the figure. The paragraph needs to be worded correctly.
· Figure 3c (RDA ordering analysis) is not mentioned anywhere in the text, only figures 3a and 3b are mentioned. A text describing what is presented in figure 3c should be included.
· In section 2.5 of materials and methods, the protocol used for the amplification and sequencing of bacterial DNA is mentioned, however, no mention is made of the results of the findings obtained, only the results for fungi are presented.

Author Response
In section 2.1 please use abbreviations such as kg, g, mg.
We have followed your suggestion and used abbreviations such as kg, g, and mg in Section 2.1 to ensure consistency and conciseness.
In section 2.2 include information on the origin of the additives used. In section 2.6 I suggest omitting the paragraph from lines 161 to 163 which talks about the link to the QIIME 2 documentation.
We have added information on the origin of the additives used in Section 2.2.
On line 168 correct fasttree2 to FastTree2
We have corrected "fasttree2" to "FastTree2" on line 168 to ensure the accuracy and consistency of the terminology.
On lines 184 and 185 correct "Funguild were....." to "FUNGuild was...."
We have corrected "Funguild were……" to "FUNGuild was……" on lines 184 and 185 to ensure grammatical and terminological accuracy.
In the paragraph from lines 232 to 239, the results of the effect of the treatments on the correlation of fungal and bacterial communities are mentioned and reference is made to figures 2a, 2b and 2c to show the increase in the relative abundance of bacterial genera (figure 2a), the significant increase in pathogenic bacteria in two of the treatments (figure 2b) and the increase in the relative abundance of the Talaromyces genus in the FLA treatment (figure 2c), however, the figures only show the taxonomic relationship of 15 fungal genera (figure 2a), the distribution of 4 genera of fungi in the different treatments (figure 2b) and the abundance of 4 genera of fungi in the different treatments (figure 2c). Therefore, the description of the results in the paragraph does not match the figures to which they refer. There is no clarity between the paragraph and the figure. The paragraph needs to be worded correctly.
We have revised the paragraph from lines 232 to 239 to remove references to bacterial genera and ensure that the description matches the fungal genera shown in Figures 2a, 2b, and 2c.
Figure 3c (RDA ordering analysis) is not mentioned anywhere in the text, only figures 3a and 3b are mentioned. A text describing what is presented in figure 3c should be included.
We have included a description of Figure 3c (RDA ordering analysis) in the text, explaining what is presented and its significance to the study results.
In section 2.5 of materials and methods, the protocol used for the amplification and sequencing of bacterial DNA is mentioned, however, no mention is made of the results of the findings obtained, only the results for fungi are presented.
The reference to bacterial sequencing was removed.
Reviewer 2 Report (Previous Reviewer 3)
Lines 65-66: I recommend to change the phrase “the pathogenicity of fusarium by stimulating the pathogen to produce some enzymes that play a pathogenic role in infecting the host” by the “the pathogenicity of Fusarium cells by stimulating them to produce enzymes playing a notable role in host infecting”.
Lines 131, 198: the “physic-chemical” should be hanged by “physico-chemical”
Tables 1 and 2: Please, leave only one significant digit after the decimal point everywhere. It will be correct. Same should be done with all calculated percentages given in the text.
Figures 1, 2, 3 and 4 should be given after their mentioning in the text.
The Titles for Figures 1, 2, 3 and 4 should be added to the text! Now they are absent. It is strange that all authors agreed with such revised version of the article!
Line 228: Fusarium should be in italic font.
Lines 230, 233, 234,236, 239, 246?351: Please, see “bacterial genera”. THERE ARE NO BACTERIA at ALL! It is very strange that all authors agreed with such revised version of the article! Please, clean the text carefully.
Please, see the template for the preparation of the article for the JOF. The numeration of the sections inside the article should be done in accordance with the recommendations of the Journal. For example, the authors should havefollowing numerations of the article sections: 1. Inrtroduction, 2. Materials and methods; 3. Results, 3.1. The effects of exogenous material addition on soil factors, etc., 4. Discussion, 4.1. The effect on soil microbial community diversity, etc.
Line 292: Please, change the title of subsection “The effect on soil microbial community diversity” – please, add the effect of what of soil is discussed here
Line 325: Please, add which impact on microbial structure is discussed here.
List of references: Previously I recommended to write all names of microorganisms in italic font even in the List with References and carefully check that. The response of the authors was following: “Thank you very much for your detailed advice. As for the reference, it has been perfected.” However, that was not done! Please, see:
- Ref#2 Penicillium italicum
- Ref#3 Phytophthora sojae
- Ref#4 Talaromyces flavus, Verticillium dahlia
- Ref#13 Panax notoginseng
- Ref#14 Fusarium oxysporum
- Ref#16 Kandelia obovata
- etc.
Please, do it really, not by words in the Cover Letter.
Lines 65-66: I recommend to change the phrase “the pathogenicity of fusarium by stimulating the pathogen to produce some enzymes that play a pathogenic role in infecting the host” by the “the pathogenicity of Fusarium cells by stimulating them to produce enzymes playing a notable role in host infecting”.
Lines 131, 198: the “physic-chemical” should be hanged by “physico-chemical”
Tables 1 and 2: Please, leave only one significant digit after the decimal point everywhere. It will be correct. Same should be done with all calculated percentages given in the text.
Figures 1, 2, 3 and 4 should be given after their mentioning in the text.
The Titles for Figures 1, 2, 3 and 4 should be added to the text! Now they are absent. It is strange that all authors agreed with such revised version of the article!
Line 228: Fusarium should be in italic font.
Lines 230, 233, 234,236, 239, 246?351: Please, see “bacterial genera”. THERE ARE NO BACTERIA at ALL! It is very strange that all authors agreed with such revised version of the article! Please, clean the text carefully.
Please, see the template for the preparation of the article for the JOF. The numeration of the sections inside the article should be done in accordance with the recommendations of the Journal. For example, the authors should havefollowing numerations of the article sections: 1. Inrtroduction, 2. Materials and methods; 3. Results, 3.1. The effects of exogenous material addition on soil factors, etc., 4. Discussion, 4.1. The effect on soil microbial community diversity, etc.
Line 292: Please, change the title of subsection “The effect on soil microbial community diversity” – please, add the effect of what of soil is discussed here
Line 325: Please, add which impact on microbial structure is discussed here.
List of references: Previously I recommended to write all names of microorganisms in italic font even in the List with References and carefully check that. The response of the authors was following: “Thank you very much for your detailed advice. As for the reference, it has been perfected.” However, that was not done! Please, see:
- Ref#2 Penicillium italicum
- Ref#3 Phytophthora sojae
- Ref#4 Talaromyces flavus, Verticillium dahlia
- Ref#13 Panax notoginseng
- Ref#14 Fusarium oxysporum
- Ref#16 Kandelia obovata
- etc.
- Please, do it really, not by words.
Author Response
Lines 65-66: I recommend to change the phrase “the pathogenicity of fusarium by stimulating the pathogen to produce some enzymes that play a pathogenic role in infecting the host” by the “the pathogenicity of Fusarium cells by stimulating them to produce enzymes playing a notable role in host infecting”.
We have revised the wording on lines 65-66 from "the pathogenicity of fusarium by stimulating the pathogen to produce some enzymes that play a pathogenic role in infecting the host" to "the pathogenicity of Fusarium cells by stimulating them to produce enzymes playing a notable role in host infecting" as you suggested.
Lines 131, 198: the “physic-chemical” should be hanged by “physico-chemical”
We have corrected "physic-chemical" to "physico-chemical" on lines 131 and 198 to ensure terminological accuracy.
Tables 1 and 2: Please, leave only one significant digit after the decimal point everywhere. It will be correct. Same should be done with all calculated percentages given in the text.
We have adjusted all data in Tables 1 and 2 to retain only one significant digit after the decimal point and have done the same for all calculated percentages in the text.
Figures 1, 2, 3 and 4 should be given after their mentioning in the text.
We have added titles for Figures 1, 2, 3, and 4 in the text to ensure completeness and readability.
The Titles for Figures 1, 2, 3 and 4 should be added to the text! Now they are absent. It is strange that all authors agreed with such revised version of the article!
We have added titles for Figures 1, 2, 3, and 4 in the text to ensure completeness and readability.
Line 228: Fusarium should be in italic font.
We have changed "Fusarium" on line 228 to italic font to conform to academic standards.
Lines 230, 233, 234,236, 239, 246?351: Please, see “bacterial genera”. THERE ARE NO BACTERIA at ALL! It is very strange that all authors agreed with such revised version of the article! Please, clean the text carefully.
We have carefully reviewed and cleaned the text, removing all erroneous references to bacterial genera to ensure that the text only discusses fungal-related content.
Please, see the template for the preparation of the article for the JOF. The numeration of the sections inside the article should be done in accordance with the recommendations of the Journal. For example, the authors should havefollowing numerations of the article sections: 1. Inrtroduction, 2. Materials and methods; 3. Results, 3.1. The effects of exogenous material addition on soil factors, etc., 4. Discussion, 4.1. The effect on soil microbial community diversity, etc.
We have adjusted the section numbering of the article in accordance with the template for the Journal of Fungi (JOF) to ensure compliance with the journal's formatting requirements. For example, the section numbering has been adjusted to: 1. Introduction, 2. Materials and Methods, 3. Results, 3.1. The effects of exogenous material addition on soil factors, 4. Discussion, 4.1. The effect on soil microbial community diversity, etc.
Line 292: Please, change the title of subsection “The effect on soil microbial community diversity” – please, add the effect of what of soil is discussed here
We have amended the title of line 292 to make it clear that the discussion focuses on the reduction of soil microbial community diversity due to the addition of foreign substances.
Line 325: Please, add which impact on microbial structure is discussed here.
We added a specific description in line 325 to clarify that the discussion focused on the effect of exogenous substance addition on fungal community composition.
List of references: Previously I recommended to write all names of microorganisms in italic font even in the List with References and carefully check that. The response of the authors was following: “Thank you very much for your detailed advice. As for the reference, it has been perfected.” However, that was not done!
We sincerely apologize for the oversight in our previous revision. Upon re-examining all references, we have implemented the comprehensive corrections. The complete reference list has been updated in the manuscript. We deeply appreciate your meticulous review that helped us strengthen this critical scholarly element.
Round 2
Reviewer 1 Report (Previous Reviewer 2)
I have carefully read the modifications made to the manuscript and consider that they are appropriate and the article can be published in its current form. I acknowledge the effort made by the authors to modify the manuscript and communicate the findings of the work carried out which will be of interest and relevance to the scientific community.
I have no detail comments. All modifications have been made
Author Response
请参阅附件。
Reviewer 2 Report (Previous Reviewer 3)
There are NO mentioning of Tables A2, A3, and A4 as well as Figures A1 and A2 given in the Appendix in the main text of the article! All of them should be added.
Line 65: Fusarium should be in italic font.
Line 257: See the title “3.4. Relationship between soil properties and fungal communities and pathogenic bacteria”. Again, and again same thing… Where are BACTERIA here??? There are no them at all!
Line 282: The mentioning of the Figure 3 is here, whereas the Figure 3 as itself is located in Line 259. I need to repeat for the authors that all “Figures should be given after their mentioning in the text”.
All references should be performed in format suggested by the MDPI Journal of Fungi (please, see template): Author 1, A.B.; Author 2, C.D. Title of the article. Abbreviated Journal Name Year, Volume, page range. Title and volume of the journal should be given in italic font, and year should be given in bold font. All references in this article should be prepared by the authors in accordance with the rules existing for all authors. This is obligation of the authors but of the editors. So, I notably recommend the authors to do this extensive work by themselves.
Reference No.46: The letters in the title of the article should be small as they are in other references.
There are NO mentioning of Tables A2, A3, and A4 as well as Figures A1 and A2 given in the Appendix in the main text of the article! All of them should be added.
Line 65: Fusarium should be in italic font.
Line 257: See the title “3.4. Relationship between soil properties and fungal communities and pathogenic bacteria”. Again, and again same thing… Where are BACTERIA here??? There are no them at all!
Line 282: The mentioning of the Figure 3 is here, whereas the Figure 3 as itself is located in Line 259. I need to repeat for the authors that all “Figures should be given after their mentioning in the text”.
All references should be performed in format suggested by the MDPI Journal of Fungi (please, see template): Author 1, A.B.; Author 2, C.D. Title of the article. Abbreviated Journal Name Year, Volume, page range. Title and volume of the journal should be given in italic font, and year should be given in bold font. All references in this article should be prepared by the authors in accordance with the rules existing for all authors. This is obligation of the authors but of the editors. So, I notably recommend the authors to do this extensive work by themselves.
Reference No.46: The letters in the title of the article should be small as they are in other references.
Author Response
- There are NO mentioning of Tables A2, A3, and A4 as well as Figures A1 and A2 given in the Appendix in the main text of the article! All of them should be added.
Tables A2, A3, and A4 are all referenced in the text. After careful consideration, Figure A1 was deleted, Figure A2 was retained, and adjustments were made to the original text. The table's original wording will not be altered.
- Line 65: Fusarium should be in italic font.
Changed "Fusarium" to italic font where it was previously mentioned in line 65.
- Line 257: See the title “3.4. Relationship between soil properties and fungal communities and pathogenic bacteria”. Again, and again same thing… Where are BACTERIA here??? There are no them at all!
Ensured that all mentions of "Bacteria" or "Bacterias" were included and corrected.
- Line 282: The mentioning of the Figure 3 is here, whereas the Figure 3 as itself is located in Line 259. I need to repeat for the authors that all “Figures should be given after their mentioning in the text”.
Ensured that all figures mentioned in the text are properly referenced and placed before their mention. Rechecked figure captions for clarity and accuracy.
- All references should be performed in format suggested by the MDPI Journal of Fungi (please, see template): Author 1, A.B.; Author 2, C.D. Title of the article. Abbreviated Journal Name Year, Volume, page range. Title and volume of the journal should be given in italic font, and year should be given in bold font. All references in this article should be prepared by the authors in accordance with the rules existing for all authors. This is obligation of the authors but of the editors. So, I notably recommend the authors to do this extensive work by themselves.
Reformatted all references according to the MDPI Journal of Fungi template, ensuring consistency in author names, journal titles, volume numbers, and publication years.
- Reference No.46: The letters in the title of the article should be small as they are in other references.
Adjusted the title of Reference No. 46 to follow standard case sensitivity rules.
Round 3
Reviewer 2 Report (Previous Reviewer 3)
Please, place ALL figures AFTER their first mentioning in the text. The new parts of the article can not begin from figures. Some text should be given in the beginning of each section instead of figures but with their mentioning.
Please, place ALL figures AFTER their first mentioning in the text. The new parts of the article can not begin from figures. Some text should be given in the beginning of each section instead of figures but with their mentioning.
This manuscript is a resubmission of an earlier submission. The following is a list of the peer review reports and author responses from that submission.
Round 1
Reviewer 1 Report
Dear Authors,
The manuscript presents a very interesting investigation, describing the results in detail. However, it is challenging to conduct a thorough review due to the lack of figures and tables presenting the results. Please include the tables and figures in the manuscript to facilitate revision.
Additionally, it is worth revising and clarifying the following information:
Lines 116-118: Did the applied methanol negatively affect the soil microbiota?
Lines 122-124: Weren't diluted hydrochloric acid and ammonia to maintain a constant soil pH harmful to the microbiota?
Reviewer 2 Report
The manuscript does not contain tables or figures, so I recommend sending the complete information so that it can be properly evaluated. Only three tables (A1, A2 and A3) and one figure (A1) were attached as complementary files. It is essential to include this information since the topic addressed and the results are of great interest and relevance.
Please include figures 1a to 1d, 2a to 2c, 3a, 3b and 4a to 4c. Also tables 1 and 2.
Sections 2.1, 2.3, 2.4, 2.5, 2.6 and 2.7 should be modified as they are practically identical to articles by the same authors in different journals and may be misinterpreted as plagiarism.
Reviewer 3 Report
I would like to draw attention of the authors to the fact that there are no real results in the article. Two tables and 4 figures mentioned in the text are missing! So, there are no to review in the part of Results.
The article should be rejected now because of its incompleteness.
In addition, if the authors will insert tables and figures into the article and forward their paper again to the same journal, I recommend the authors to use template to prepare the manuscript for the second submission.
I suggest the authors to take into account the following recommendations:
Introduction
Line 65: There is the phrase “the production of pathogenic enzymes”. Which enzymes are “pathogenic” and what is the difference between pathogenic and non-pathogenic enzymes? I recommend to rephrase the sentence.
Lines 68,69: I recommend to replace the phrase “organic acid compounds” by “organic acids”.
Lines 371-372: Same recommendation concerns the phrase “organic acid substances”. It is more correct to use “organic acids”.
Line 77: “spp.” should not be in italic font.
References
There are no DOI in the references at all, and this information should be added to each reference if it is possible.
There are no page numbers in the 16 references: No. 2, 3, 5, 7, 10, 13, 14, 17, 18, 20, 21, 22, 50, 51, 59, 65.
There are no volume and pages numbers in the reference No. 39.
All names of microorganisms should be in italic font even in the List with References. It should be carefully checked.
Supplementary Materials
Table A2. I recommend to give the explanation of the abbreviations CK, PA, FLA, OA here or in the Table A1. Additionally, please note, what was accepted here as 100% in the calculation of Relative abundance.
Table A3. I recommend to add the explanation of all abbreviations mentioned here as table footnotes.
Figure A1: There is the phrase in the capture of the Figure: “pathogenic bacteria”. There are no bacteria at all! Only fungi!
I would like to draw attention of the authors to the fact that there are no real results in the article. Two tables and 4 figures mentioned in the text are missing! So, there are no to review in the part of Results.
The article should be rejected now because of its incompleteness.
In addition, if the authors will insert tables and figures into the article and forward their paper again to the same journal, I recommend the authors to use template to prepare the manuscript for the second submission.
I suggest the authors to take into account the following recommendations:
Introduction
Line 65: There is the phrase “the production of pathogenic enzymes”. Which enzymes are “pathogenic” and what is the difference between pathogenic and non-pathogenic enzymes? I recommend to rephrase the sentence.
Lines 68,69: I recommend to replace the phrase “organic acid compounds” by “organic acids”.
Lines 371-372: Same recommendation concerns the phrase “organic acid substances”. It is more correct to use “organic acids”.
Line 77: “spp.” should not be in italic font.
References
There are no DOI in the references at all, and this information should be added to each reference if it is possible.
There are no page numbers in the 16 references: No. 2, 3, 5, 7, 10, 13, 14, 17, 18, 20, 21, 22, 50, 51, 59, 65.
There are no volume and pages numbers in the reference No. 39.
All names of microorganisms should be in italic font even in the List with References. It should be carefully checked.
Supplementary Materials
Table A2. I recommend to give the explanation of the abbreviations CK, PA, FLA, OA here or in the Table A1. Additionally, please note, what was accepted here as 100% in the calculation of Relative abundance.
Table A3. I recommend to add the explanation of all abbreviations mentioned here as table footnotes.
Figure A1: There is the phrase in the capture of the Figure: “pathogenic bacteria”. There are no bacteria at all! Only fungi!